# Functional Reconstruction of Denervated Muscle by Xenotransplantation of Neural Cells from Porcine to Rat

**DOI:** 10.3390/ijms23158773

**Published:** 2022-08-07

**Authors:** Sota Saeki, Katsuhiro Tokutake, Masaki Takasu, Shigeru Kurimoto, Yuta Asami, Keiko Onaka, Masaomi Saeki, Hitoshi Hirata

**Affiliations:** 1Department of Human Enhancement and Hand Surgery, Nagoya University Graduate School of Medicine, 65 Tsurumai-cho, Showa-ku, Nagoya 466-8550, Japan; 2Department of Veterinary Medicine, Faculty of Applied Biological Sciences, Gifu University, Yanagido 1-1, Yanagido, Gifu 501-1193, Japan

**Keywords:** xenotransplantation, neural stem cell, regenerated axons, functional recovery, muscle atrophy, electrical stimulation

## Abstract

Neural cell transplantation targeting peripheral nerves is a potential treatment regime for denervated muscle atrophy. This study aimed to develop a new therapeutic technique for intractable muscle atrophy by the xenotransplantation of neural stem cells derived from pig fetuses into peripheral nerves. In this study, we created a denervation model using neurotomy in nude rats and transplanted pig-fetus-derived neural stem cells into the cut nerve stump. Three months after transplantation, the survival of neural cells, the number and area of regenerated axons, and the degree of functional recovery by electrical stimulation of peripheral nerves were compared among the gestational ages (E 22, E 27, E 45) of the pigs. Transplanted neural cells were engrafted at all ages. Functional recovery by electric stimulation was observed at age E 22 and E 27. This study shows that the xenotransplantation of fetal porcine neural stem cells can restore denervated muscle function. When combined with medical engineering, this technology can help in developing a new therapy for paralysis.

## 1. Introduction

Extensive research has been carried out to develop treatments for neurological disorders such as amyotrophic lateral sclerosis and spinal cord injury; however, no fundamental treatment has been established. Cell therapies for the central nervous system, including the spinal cord, have also been studied [1,2,3]. Although stem cell transplantation is desirable to compensate for motor paralysis resulting from denervation, it is necessary for motor neurons transplanted into the spinal cord to extend the axon a considerable distance to the target muscle in order to replace the function of lower motor neurons. Given the difficulty in recovering denervated muscles due to prolonged reinnervation time [4,5], it is quite difficult for lower motor neuron diseases to complement motor neurons post-stem-cell transplantation into the central nervous system.

With the development of medical engineering such as brain–computer interfaces (BCIs) [6,7,8], recovery of the motor function of patients with paralysis using functional electrical stimulation has also been attempted. However, this technology is based on innervated skeletal muscles. Attempts have been made to supplement muscle strength by placing electrodes in the skeletal muscles for lower motor neuron disease, but no effect was observed after denervation [9]. In muscles that are losing innervation, neuromuscular junctions and muscle fibers gradually degenerate, and the response to electrical stimulation decreases, resulting in a loss of muscle contraction [10].

To treat atrophic muscles after denervation, neural stem cells have been transplanted into peripheral nerves near target muscles [11,12,13,14]. The distance to the target muscle is short; therefore, the time required for reinnervation is reduced, and efficient muscle force recovery can be expected. In addition, because only hundreds or thousands of motor units need to be reconstructed, the transplantation of cells into peripheral nerves requires fewer transplanted cells. Thus, we have been working to transplant spinal-cord-derived cells of fetal rats into transected peripheral nerves to reconstruct the function. Additionally, we succeeded in controlling the regenerated muscle via functional electrical stimulation [14]. Sensory nerves as well as motor nerves have been successfully regenerated by grafting dorsal root ganglia [15,16].

One of the challenges of future clinical applications of this technology is to obtain donor cells that function as neurons in peripheral nerves and are resistant to tumorigenesis and immune rejection. To date, in addition to rat embryonic neural stem cells, mouse embryonic stem (ES) cells [13,17], mouse-induced pluripotent stem (iPS) cells [18], and human iPS cells [19] have been transplanted into peripheral nerves. Ethical concerns are associated with transplanted cells of allogeneic fetal origin. Additionally, tumorigenesis remains an issue with ES cells [20]. iPS cells have been accepted and developed in order to avoid these ethical problems. However, reproducing the in vitro developmental process to facilitate cellular differentiation through highly complex interactions and obtain cells with functions equivalent to those of lower motor neurons is not easy. The histological regeneration of neuromuscular junctions was demonstrated by human iPS cell transplantation, but the functional recovery was not demonstrated by electrophysiological evaluation [19]. The induction of iPS cells differentiation into sensory neurons has also been attempted [21], but it is difficult to generate functional sensory neurons in vivo.

Therefore, we focused on porcine-fetal-derived neural stem cells for xenotransplantation, whose outcomes have improved considerably due to the recent developments in genome editing technology [22]. Furthermore, with the rapid advancements in immunosuppressive therapy [23], the expectation that xenotransplantation can be used in clinical settings is rapidly increasing. We aimed to develop a new therapeutic technique for intractable muscle atrophy caused by lower motor neuron disease by transplanting neural stem cells derived from pig fetuses into peripheral nerves.

One of the most important factors to consider when using fetal cells for transplantation is the age of the fetus. Research on rodent neurons has shown that the 14th day of gestation is the optimum time for maximal neuronal differentiation [24]. It has been found that if neural cells from this period are harvested and transplanted into peripheral nerves, the transplanted stem cells can differentiate into motor neurons and functionally re-innervate denervating muscles [11,12,14]. However, studies on pigs are still sparse, and the timeline of porcine neuronal differentiation is unknown. Additionally, neural stem cells derived from xenogeneic fetuses have not been reported to differentiate into motor neurons and form functional neuromuscular junctions.

Therefore, the aim of this study was to investigate whether porcine fetal neural stem cells can engraft in peripheral nerves of rats and regenerate functional neuromuscular junctions. Furthermore, we assessed the transplantation of cells from fetuses harvested at different times with different degrees of neuronal differentiation. We believe that this study can contribute to the future development of xenotransplantation.

## 2. Results

### 2.1. Effective Engraftment and Neurogenesis Post-Transplantation

Immunocytochemistry revealed the presence of β3-tubulin-positive neurons in all transplantation groups, indicating the engraftment of neurons in the peripheral nerves. A small number of glial fibrillary acidic protein (GFAP)-positive astrocytes were seen in group E 45, but not in groups E 22 and E 27 (Figure 1a–c). The proportions of β3-tubulin-positive neurons were 29.1 ± 3.2%, 57.1 ± 5.9%, and 16.1 ± 1.5% in E 22, E 27, and E 45, respectively. The proportions of β3-tubulin-positive neurons were higher in E 22 and E 27 as compared with that in E 45, with significant differences between E 22 and E 27 (*p* < 0.01) and E 27 and E 45 (*p* < 0.01). Neurogenesis is characterized by the differentiation of neurons followed by astrocytes and subsequently oligodendrocytes [24,25]. The percentages of neurons indicate that the peak of neurogenesis is close to day 27 of gestation.

### 2.2. Electrophysiological Evaluation

The mean amplitudes of the complex muscle action potential (CMAP) in the E 22 and E 27 groups were 1.47 ± 0.97 mV and 3.64 ± 2.85 mV, respectively (Figure 2a). The amplitude was 0 mV in all rats in the E 45 and control groups. There was no significant difference between any of the groups.

Direct electrical stimulation of the peroneal nerve caused ankle dorsiflexion in four of eight rats in the E 22 group and in four of five rats in the E 27 group. The mean ankle dorsiflexion angles were 9.88 ± 5.42° and 17.2 ± 12.29° in the E 22 and E 27 groups, respectively (Figure 2b). The difference between these groups was not significant.

### 2.3. Tissue Analysis

#### 2.3.1. Effects of Preventing Muscle Atrophy

The tibialis anterior muscle wet weight (as a percentage of body weight [% BW]) in the E 22, E 27, E 45, and surgical control groups was 0.0425 ± 0.0029%, 0.0544 ± 0.006%, 0.0431 ± 0.0019%, and 0.0413 ± 0.0025%, respectively. There was no significant difference between all groups (Figure 3).

#### 2.3.2. Histological Evaluation of the Distal Nerve Stump

The stump of the peroneal nerve into which the cell transplantation was carried out was swollen (Figure 4a). In contrast, there was no enlargement of the stump in the surgical control group (Figure 4b). Fluorescent immunostaining revealed β3-tubulin-positive neurons and GFAP-positive astrocytes in the peripheral nerves at the site of cell transplantation in all groups (Figure 4c). These results indicated that porcine neurons and glial cells survived in the peripheral nerves of rats 12 weeks after transplantation.

#### 2.3.3. Evaluation of the Neuromuscular Junction

In groups E 22 and E 27, regenerated axons reached the motor end plates, as indicated by α-bungarotoxin labeling of the acetylcholine receptor cluster (Figure 5a). No axons reached the motor end plates in the E 45 and control groups (Figure 5b). In group E 45, few neurons survived, but axons did not extend to form functional motor units with the denervated muscles.

#### 2.3.4. Evaluation of Retrograde Tracing

Some of the cholinergic neurons in the distal nerve stump were stained by the retrograde tracer in groups E 22 and E 27 (Figure 6). Cholinergic neurons in the peripheral nerve reached the denervated muscle and formed the neuromuscular junction. The fusiform structure of the distal nerve stump was also found to be a ganglion containing cholinergic neurons.

#### 2.3.5. Evaluation of Regenerated Myelinated Axons

Toluidine blue staining confirmed the presence of myelinated axons in the peroneal nerve in all the transplantation groups (Figure 7a–c). No myelinated axons were observed in the peroneal nerve in the control group (Figure 7d). Thus, the pig fetal-derived neurons engrafted into the peripheral nerves of nude rats extended their axons in all the groups. The number of myelinated axons in groups E 22, E 27, and E 45 was 407.63 ± 104.38, 459.8 ± 122.5, and 192.5 ± 112.11, respectively. The E 27 group had the highest number; however, the differences between the groups were not significant (Figure 7e). The mean cross-sectional area of the myelinated axons in groups E 22, E 27, and E 45 was 6.37 ± 0.13 μm^2^, 10.60 ± 0.35 μm^2^, and 7.02 ± 0.33 μm^2^, respectively. The mean cross-sectional area of the myelinated axons was significantly lower in group E 22 than in groups E 27 (*p* < 0.01) and E 45 (*p* < 0.01) (Figure 7f). The distribution of myelinated axon diameter in each group did not differ among the groups (Figure 7g).

## 3. Discussion

Our findings indicated that in all transplantation groups (E 22, E 27, and E 45), porcine neural cells engrafted in the peripheral nerves of nude rats and extended their axons, according to histological evaluation. Choline acetyltransferase (ChAT) staining and retrograde tracing showed that the transplanted cells differentiated into motor neurons and formed neuromuscular junctions in groups E 22 and E 27. Furthermore, electrophysiological evaluation revealed that motor function was restored in the E 22 and E 27 groups. The percentages of β3-tubulin-positive neurons was the highest in E 27 group, and the mean axon area in the E 27 group was significantly larger than that in the E 22 group, suggesting that the E 27 group is an appropriate donor for porcine neural cell transplantation with the best motor function recovery.

Erb et al. were the first to report the use of rat fetal neural cells for transplantation into peripheral nerves which were disconnected from the central nervous system [11]. Later, motor neuron progenitor cells derived from mouse ES cells expressing channelrhodopsin−2 were transplanted into the peripheral nerve trunk, where muscle contraction was assessed by light control [13]. Although there have been studies using mouse [26] and human iPS cells [19], this is the first study in which fetal porcine neural cells were transplanted into xenogeneic peripheral nerves.

Although immunological problems and complications of the coagulation system have long been considered as problems in xenotransplantation, many issues have been addressed by the development of genetically modified pigs [26]. In addition, endogenous retroviruses cause the most problematic infectious diseases in pigs [27], but the CRISPR-Cas9 technique was developed to inactivate all retroviral genes [22]. Cell transplantation, as in this study, has fewer complications associated with immune and coagulation systems than that in the organ transplantation, and it is easier to control infection using fetal cells. Therefore, its clinical applications could be considered in the near future. In this regard, porcine islets have already been transplanted into patients with type 1 diabetes [28]. Similarly, neurons derived from fetal pig brains have also been transplanted into patients with Parkinson’s disease [29]. Moreover, preclinical studies have been conducted where neurons derived from genetically modified fetal pig brains have been transplanted into monkey models of Parkinson’s disease [30].

The optimal time point for harvesting fetal spinal-cord-derived neurons in rats is the 14th day of gestation [24], when the differentiation of most neurons is initiated. Fetal dopaminergic neurons in rats have also been optimally harvested for transplantation around day 14 of gestation [31,32]. The fetal porcine dopaminergic neurons are best collected on days 26 to 27 of gestation [33]. Based on these reports, the optimal time for harvesting fetal porcine spinal cords can be predicted to be around 27 days, which is consistent with the present experimental results, thus suggesting that the differentiation to the neuron has advanced the most by E 27. Conversely, the results of electrophysiological and histological evaluations in the E 45 group indicated that although the neurons engrafted and extended axons, they did not form functional neuromuscular junctions.

This study has several limitations. First, the sample size was small. A larger sample could have provided a better understanding and defined the differences in transplant outcomes by gestational age. However, the present experimental results show that motor neurons derived from xenogeneic fetuses engraft in peripheral nerves and form functional neuromuscular junctions. In addition, the results of cell culture and electrophysiological evaluation showed that 45 days of gestation was too late for fetuses to be harvested, and this finding was very important for further study. Second, it is not clear how the transplanted porcine cells differentiate into motor neurons, form ectopic ganglia, and re-form neuromuscular junctions in denervated muscles. From the research on neural stem cells derived from allogeneic fetuses thus far, it is considered that not only motor neurons, but glial cells differentiated from neural stem cells are also important for the functional recovery of denervated muscle [34,35,36]. In this regard, the detailed mechanism needs to be clarified in future studies. Third, transplant conditions need to be further examined. Deshpande et al. demonstrated that the transplantation of myelin inhibitors and administration of the neurotrophic factor to peripheral nerves promote axonal growth towards the muscle [37]. Several studies have investigated factors that improve the survival of motor neurons in peripheral nerves [12,38,39]. Further experiments are necessary to examine and optimize the conditions of the graft site for future clinical applications. Additionally, reinnervated muscles are not connected to the brain, and hence, cannot be contracted voluntarily. However, recent developments in BCI [6,7,8] have made it possible to convert signals from the brain into electrical signals that are transmitted to the muscles through electrical neural bypasses and help to control movement by electrical stimulation. In the future, by combining our developed technology with BCI, it will be possible to achieve a new treatment for paralysis.

In conclusion, the present study showed that xenogeneic neural cell transplantation to peripheral nerves induces the reinnervation of denervated muscles. Therefore, this functional regeneration technology via neural cell transplantation can provide a new therapy for paralysis, when combined with medical engineering.

## 4. Materials and Methods

### 4.1. Animals

Twenty-two (7-week-old) nude rats (F 344/NJcl-rnu/rnu; CLEA Japan Inc., Tokyo, Japan) were used as denervation model rats through unilateral transection of the peroneal nerve at the femoral level. To prevent axonal regeneration by proximal axonal extension, the proximal portion of the peroneal nerve was ligated with 5–0 nylon and sutured to the gluteal muscle (Figure 8c). For the same reason, the distal stump of the peroneal nerve was ligated using 5–0 nylon. To obtain pig fetuses as donors for neural cell transplantation, seven pregnant Micromini pigs (Fuji Micra Inc., Shizuoka, Japan) were prepared. Of these, 3 pigs were on gestation day 22, 2 pigs were on gestation day 27, and 2 pigs were on gestation day 45. Before beginning the experiment, all the animals were kept under conventional production conditions. All surgical procedures were performed in a sterile manner using 2% isoflurane anesthesia.

### 4.2. Cell Harvest, Preparation and Transplantation

One week after creating a model of peroneal nerve transection in recipient rats, the uterus was removed from a pregnant micromini pig under anesthesia. The dorsal root ganglia were removed from the fetal spinal cord in the uterus (Figure 8a,b), and the ventral spinal cord, including the anterior horn, was minced and placed in Hank’s balanced salt solution (Life Technologies Japan, Tokyo, Japan). The fetal spinal cord cells were then dissociated by repeated aspiration with a glass Pasteur pipette with papain-containing dissociation solution (cat 297-78101; FUJIFILM Wako Pure Chemical Corporation, Osaka, Japan). The cells were then suspended in culture medium (cat 148-09671; FUJIFILM Wako Pure Chemical Corporation, Osaka, Japan) and the dispersed cell suspension was centrifuged to collect the cells.

Recipient rats were anesthetized to expose the peroneal nerve that had been severed 1 week earlier, and 800,000 fetal spinal cord cells in 10 μL of culture medium were slowly injected into the distal stump of the peroneal nerve using a Hamilton syringe with a 30 G needle (Figure 8c). The injection site was 20 mm from the entrance to the tibialis anterior. The number of rats in each group was determined according to the number of cells obtained from pig fetuses. Cells from fetuses on day 22, 27, and 45 were transplanted into 8 rats in group E 22, 5 rats in group E 27, and 4 rats in group E 45. Cell-free medium was transplanted into 5 rats as a surgical control group (Figure 9).

### 4.3. Cell Culture

Dissected fetal spinal cord cells were distributed in uniform density (50,000 cells/well) on the chemical-coated chamber slides (Life Technologies Japan, Tokyo, Japan). These were fixed with 4% paraformaldehyde (PFA) in 0.1 M phosphate buffer after 12–24 h of culture. They were then stained with anti-β3-tubulin conjugated with Alexa Fluor 488 (Tuj1, 1:400; BioLegend, San Diego, CA, USA) and anti-GFAP antibodies [GA-5] (Cy3; 1: 400; Abcam, Cambridge, UK), and Hoechst (1: 1000; Dojindo, Kumamoto, Japan). Cells were observed under a fluorescence microscope (BZ-9000; Keyence, Osaka, Japan). Five arbitrary points were selected in each well of each transplant group, and the percentage of Tuj1-positive neurons among all cells with nuclear staining was calculated and compared.

### 4.4. Electrophysiological Evaluation

Twelve weeks after cell transplantation, electrophysiological evaluation was performed in each group. After anesthetizing with isoflurane, the distal peroneal nerve stumps of the cell transplantation sites were exposed and a bipolar stimulation electrode (UM 2-5050, Nihon Kohden, Tokyo, Japan) was placed on them. Electrical stimulation was performed with an isolator (SS-201 J, Nihon Kohden, Tokyo, Japan) connected to an electrical stimulator. The CMAP of the tibialis anterior was measured using a standard neural evoked potential recorder (MEB-9404, Neuropack S1, Nihon Kohden, Tokyo, Japan). Two stainless-steel monopolar recording electrodes (H 537 A, Nihon Kohden, Tokyo, Japan) were placed at the middle of the tibialis anterior belly and at the distal tendon. The amplitude was measured as the peak value of the waveform at the supramaximal stimulus.

Ankle joint motion was evaluated after measurement of the CMAP. Stimulation was performed using an electrical pulse (100 ms duration, 60 Hz frequency, 2.0 mA intensity, square wave) with an isolator connected to an electronic stimulator as well as CMAP. The ankle angle was calculated as the angle subtracted from the line connecting the knee and ankle and the line connecting the ankle and metatarsal head [40].

### 4.5. Animal Sacrifice

After electrophysiological evaluation, all rats were anesthetized, and the extensor hallucis longus muscle was harvested and perfused with 50 mL of 0.9% saline and then 200 mL of 4% PFA in 0.1 M phosphate buffer. The tibialis anterior muscle was then harvested and weighed immediately. The peroneal nerve was harvested and divided into proximal and distal halves. The proximal peroneal nerve, including the cell transplant site, was cryoprotected with sucrose and frozen with isopentane cooled with liquid nitrogen into suitable tissue forms containing compound (Surgipath FSC 22 Clear Frozen Section Compound, Leica Biosystems, Nussloch, Germany). The distal half of the peroneal nerve was fixed with 2% glutaraldehyde/4% PFA in phosphate buffer and embedded in the Epon.

### 4.6. Tissue Analysis

#### 4.6.1. Evaluation of the Nodule Structure in the Distal Nerve Stump

The proximal half of the frozen-embedded peroneal nerve was cut into 20 μm thick longitudinal sections, stained with Alexa Fluor 488 conjugated anti-β3-tubulin antibody (1:400; BioLegend, San Diego, CA, USA), anti-GFAP antibody [GA-5] (Cy3; 1:400; Abcam, Cambridge, UK), and Hoechst (1:1000; Dojindo, Kumamoto, Japan) and observed under a fluorescence microscope (BZ-9000; Keyence, Osaka, Japan).

#### 4.6.2. Evaluation of the Neuromuscular Junction

The harvested extensor hallucis longus muscle was clamped with a glass slide, compressed manually, and fixed in 4% PFA in 0.1 M phosphate buffer. They were stained with anti-β3-tubulin antibody (1:500; Abcam, Cambridge, UK), anti-SMI-312 (1:500; BioLegend, San Diego, CA), Alexa Fluor 594 conjugate α-bungarotoxin (α-BTX, 1:400, Life Technologies Japan, Tokyo, Japan), and observed using confocal microscopy (TiE-A1R, Nikon, Tokyo, Japan).

#### 4.6.3. Evaluation of Retrograde Tracing

Fluoro-Gold (Fluorochrome, Denver, CO, USA) was injected into the tibialis anterior muscle of E 22 and E 27 rats 1 week prior to perfusion fixation. After perfusion fixation, the proximal half of the peroneal nerve was frozen-embedded and cut in a 20 μm longitudinal section, stained with anti-β3-tubulin conjugated with Alexa Fluor 488 (1:400; BioLegend, San Diego, CA, USA) and anti-ChAT antibody (1:50; Sigma-Aldrich, St. Louis, MO, USA), and observed using confocal microscopy (TiE-A1R, Nikon, Tokyo, Japan).

#### 4.6.4. Evaluation of Regenerated Myelinated Axons

The distal half of the Epon-embedded peroneal nerve was cut into a 1 μm thick transverse section with a glass knife. Sections were stained with toluidine blue (Sigma-Aldrich, St. Louis, MO, USA) for light microscopy, and the number and area of myelinated axons were measured using ImageJ/Fiji software [41].

### 4.7. Statistical Analysis

Analysis of variance with Tukey post hoc comparisons or the Kruskal–Wallis test followed by the Bonferroni’s post hoc test was used to compare the results between groups. A comparison of muscle wet weight was performed between all groups including the control. For all the other experiments, a comparison was made between transplant groups only. All statistical analyses were conducted using the Statistical Package for Social Science (SPSS) version 28.0 software (IBM, Armonk, NY, USA). Statistical significance was set at *p* < 0.05. All data for continuous variables are presented as mean ± standard error (SE).

## Figures and Tables

**Figure 1 ijms-23-08773-f001:**
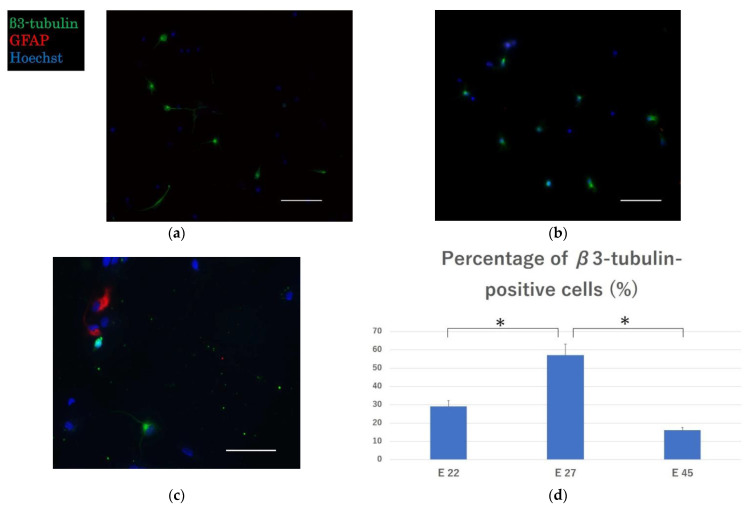
Cultured embryonic spinal cord cells from different ages. Representative image of (**a**) cells from a 22-day embryo; (**b**) cells from a 27-day embryo; (**c**) cells from a 45-day embryo; (**d**) percentage of β3-tubulin positive cells in E 22, E 27 and E 45 groups. GFAP, glial fibrillary acidic protein. Scale bar = 60 µm. * *p* < 0.05 between groups. The data are presented as mean ± standard error (SE).

**Figure 2 ijms-23-08773-f002:**
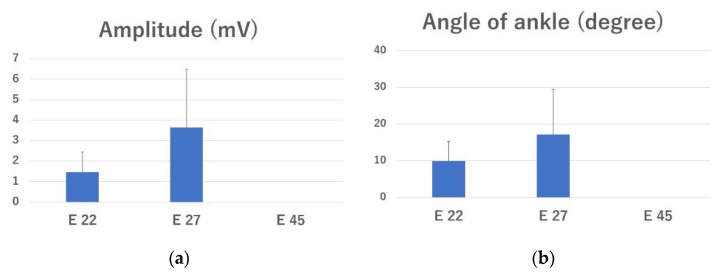
Electrophysiological evaluation. (**a**) Amplitude of compound muscle action potentials (CMAPs) recorded in the tibialis anterior muscle. (**b**) Angle of the ankle was calculated before and after electrical stimulation. The data are presented as mean ± SE.

**Figure 3 ijms-23-08773-f003:**
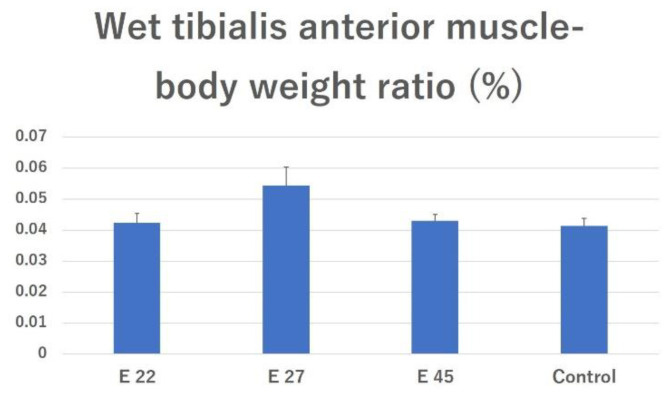
Effects of preventing muscle atrophy. Wet tibialis anterior muscle-body weight ratios of different groups were calculated and compared. The data are presented as mean ± SE.

**Figure 4 ijms-23-08773-f004:**
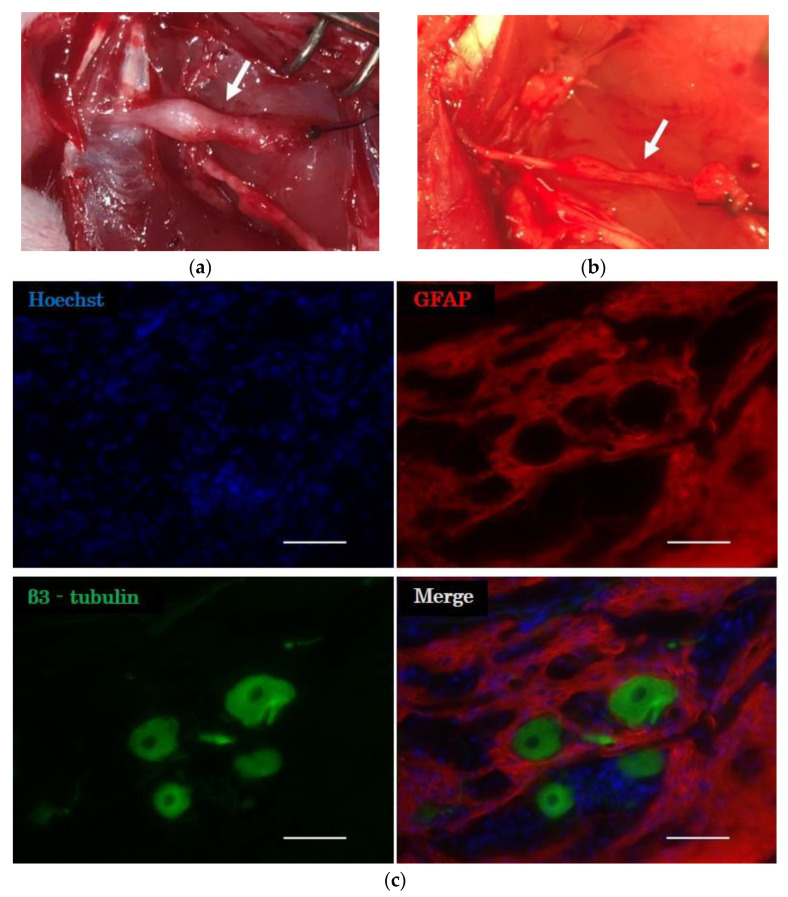
Evaluation of the nodule structure in the distal nerve stump. (**a**) Ectopic ganglion (arrow) was formed 12 weeks after cell transplantation in the E 45 group. (**b**) There was no enlargement of the stump of the peroneal nerve (arrow) in the surgical control group. (**c**) Microscopic image of the stump of the peroneal nerve from the E 27 group after immunohistochemical analysis. GFAP, glial fibrillary acidic protein. Scale bar = 50 µm.

**Figure 5 ijms-23-08773-f005:**
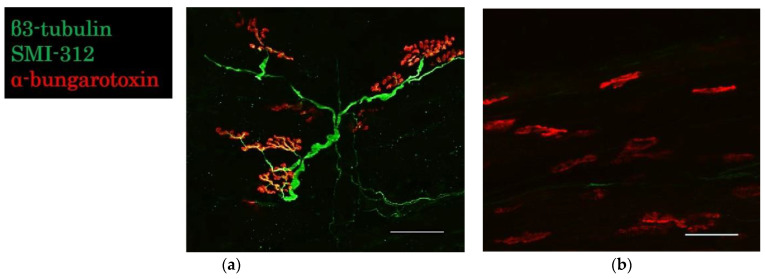
Evaluation of the neuromuscular junction after immunohistochemistry. (**a**) Microscopic image of extensor hallucis longus in the E 27 group. (**b**) Microscopic image of extensor hallucis longus in the surgical control group. Scale bar = 50 µm.

**Figure 6 ijms-23-08773-f006:**
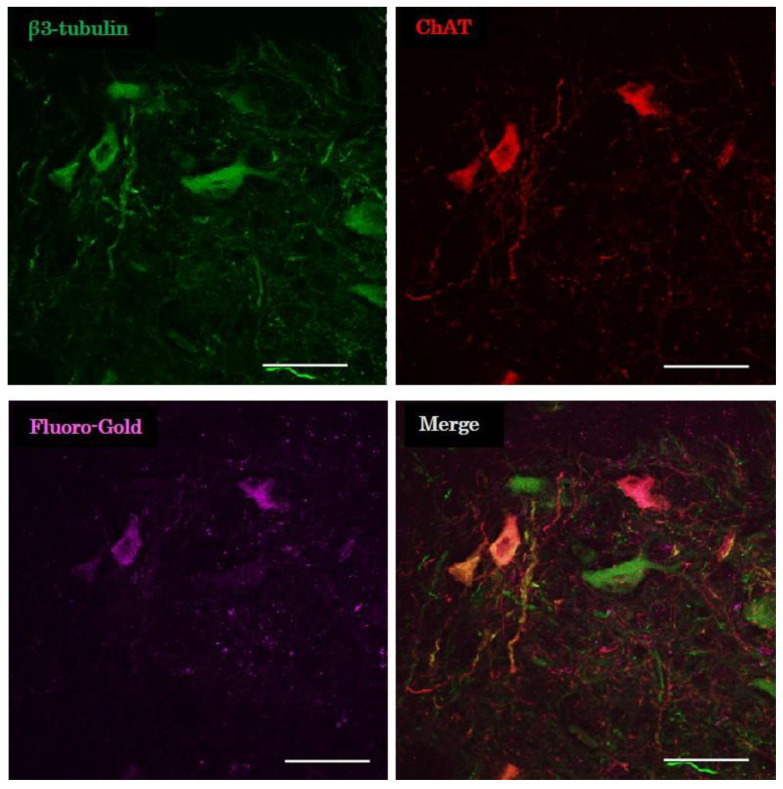
Evaluation of retrograde tracing. Microscopic image of ectopic ganglion after the injection of Fluoro-Gold and immunohistochemical staining. ChAT, choline acetyltransferase. Scale bar = 50 µm.

**Figure 7 ijms-23-08773-f007:**
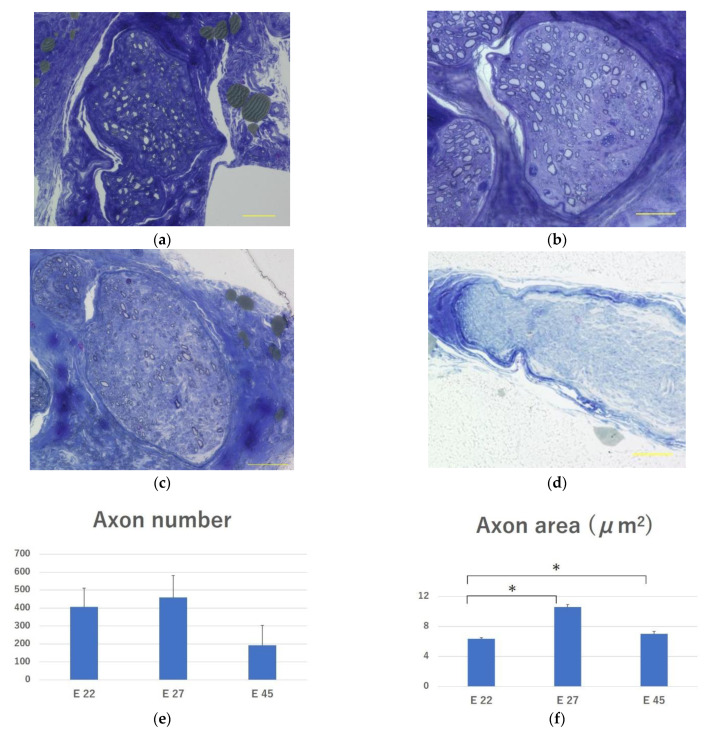
Evaluation of regenerated myelinated axons in the peroneal nerve. (**a**) Microscopic image of the peroneal nerve in the E 22, (**b**) E 27, (**c**) E 45, and (**d**) surgical control group after toluidine blue staining. (**e**) Axon number and (**f**) cross-sectional area of the myelinated axons were compared. (**g**) Distribution of myelinated axon diameter in each group (percentage of fibers (%) for each fiber diameter category). Scale bar = 50 µm. * *p* < 0.05 between groups. The data are presented as mean ± SE.

**Figure 8 ijms-23-08773-f008:**
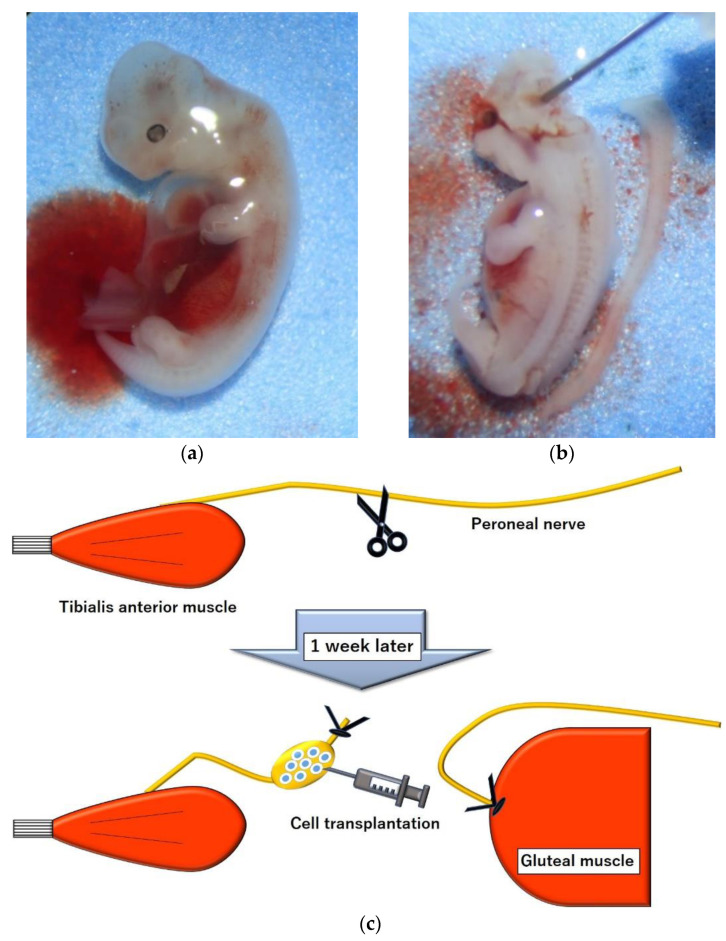
Model and cell preparations. (**a**) The 27-day-old pig embryo; (**b**) image showing harvested spinal cord from the embryo. (**c**) Diagram showing the surgical procedure for preparing the peroneal nerve of rats and injecting the medium containing neural cells into the distal stump of the peroneal nerve.

**Figure 9 ijms-23-08773-f009:**
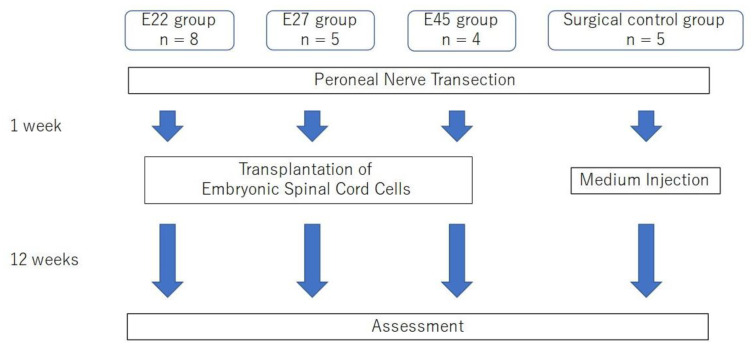
Flowchart depicting the timeline and steps for cell preparation and transplantation.

## Data Availability

The data that support the findings of this study are available from the corresponding author upon reasonable request.

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
