# Peer review of "Functional Reconstruction of Denervated Muscle by Xenotransplantation of Neural Cells from Porcine to Rat"

_ijms, 2022, doi:10.3390/ijms23158773_

Round 1
Reviewer 1 Report
In this manuscript, the authors evaluated the functional reconstruction of denervated muscle by xenotransplantation of neural cells from porcine to rat.
I would discuss in more details the findings of their work with the previous published papers on this topic, while underlining the novelty as well as the pros and cons of their technique.
Author Response
Point 1: In this manuscript, the authors evaluated the functional reconstruction of denervated muscle by xenotransplantation of neural cells from porcine to rat.
I would discuss in more details the findings of their work with the previous published papers on this topic, while underlining the novelty as well as the pros and cons of their technique.
Response 1:
Thank you for your insights and suggestions to improve the manuscript. There have been studies in which motor neurons derived from ES cells and iPS cells were transplanted into peripheral nerves to restore motor function. Induction of differentiation of iPS cells into sensory neurons has also been attempted [21], but it is difficult to generate a functional sensory neuron in vivo. We had previously induced ES cells and iPS cells into motor neurons using specific cell markers as indicators but failed to produce functional cells compared with fetal spinal cord derived cells. Furthermore, the problem of tumorigenesis of iPS cells has not been resolved. There have been no reports of transplantation of fetal spinal cord cells into xenogeneic peripheral nerves. The advantage of this method is that glial cells as well as anterior horn cells differentiated in the spinal cord can be transplanted simultaneously. In our previous report showed that both peripheral nervous system and central nervous system cells were suggested to support neuron survival and be involved in the myelination of the regenerated axon by electron microscopy [36].
We cited more published papers and we revised the sentences in the introduction lines 53-71.
“Based on the above, we have been working to transplant spinal cord derived cells of fetal rats into transected peripheral nerves to reconstruct the function. And we succeeded in controlling the regenerated muscle via functional electrical stimulation [14]. Motor nerves but also sensory nerves have been successfully regenerated by grafting dorsal root ganglia [15,16]
One of the challenges of future clinical application of this technology is to obtain do-nor cells that have functions as nerve cells of peripheral nerves and are resistant to tumorigenesis and immune rejection. To date, besides rat embryonic neural stem cells, mouse embryonic stem (ES) cells [13,17], mouse induced pluripotent stem (iPS) cells [18], and human iPS cells [19] have been transplanted into peripheral nerves. Ethical concerns are associated with transplanted cells of allogeneic fetal origin. Additionally, tumorigenesis remains an issue with ES cells [20]. The iPS cells have been accepted and developed in order to avoid these ethical problems. However, reproducing the in vitro developmental process facilitate cellular differentiation through highly complex interactions and obtain cells with functions equivalent to those of lower motor neurons is not easy. Histological regeneration of neuromuscular junctions was demonstrated by human iPS cell trans-plantation, but the functional recovery was not demonstrated by electrophysiological evaluation [19]. Induction of differentiation of iPS cells into sensory neurons has also been attempted [21], but it is difficult to generate a functional sensory neuron in vivo.”

Reviewer 2 Report
Major comments
Authors present results about peripheral nerve regeneration in a rat model after implantation of embryonal stem cells.
They methods they use are gait analysis, muscle weight, immunohistochemistry and neurophysiology. This gives some information about nerve regeneration, but measurements on protein and gene expression level are missing.
Although results seem promising, methods seems insufficient to evaluate the underlying processes of nerve regeneration in an exact manner.
It would have been of particular interest what happens in the process of embryonic stem cell engraftment, particularly regarding stem cell differentiation.
Without a more sophisticated methodology, this work does not seem suitable for publication in this journal.
Minor concerns
ESCs were taken from a different species than used in the animal model. Rat ESCs are commercially available, why didn´t authors use such a cell line? This could have helped to avoid immunogenic reactions that may have flawed the results.
Author Response
Point 1: Major comments
Authors present results about peripheral nerve regeneration in a rat model after implantation of embryonal stem cells.
They methods they use are gait analysis, muscle weight, immunohistochemistry and neurophysiology. This gives some information about nerve regeneration, but measurements on protein and gene expression level are missing.
Although results seem promising, methods seems insufficient to evaluate the underlying processes of nerve regeneration in an exact manner.
It would have been of particular interest what happens in the process of embryonic stem cell engraftment, particularly regarding stem cell differentiation.
Without a more sophisticated methodology, this work does not seem suitable for publication in this journal.
Response 1:
Thank you for your insights and suggestions to improve the manuscript. We agree with you that what happens during stem cell engraftment is important. However, as mentioned in the text as the second limitation of our study (lines 238–243 in the revised text), we did not determine how the transplanted porcine cells differentiate into motor neurons, form ectopic ganglia, and re-form neuromuscular junctions in denervated muscles. We prioritized the assessment of their survival and function in vivo instead, because we conducted a xenotransplantation experiment using pigs as donors with the aim of future clinical use. Then, this experiment showed that fetal neural stem cells can differentiate into motor neurons and function in a xenogeneic body. We should clarify the differentiation and engraftment process of the cells in future studies.
Point 2:
Minor concerns
ESCs were taken from a different species than used in the animal model. Rat ESCs are commercially available, why didn´t authors use such a cell line? This could have helped to avoid immunogenic reactions that may have flawed the results.
Response 2:
As you pointed out, allogeneic ES cells appear to be more favorable immunologically. We had previously induced ES cells and iPS cells into motor neurons using specific cell markers as indicators but failed to produce functional cells compared with fetal spinal cord derived cells. This may be because the markers indicate that the cells have differentiated into motor neurons but have not become fully functional mature motor neurons, or the glial cells transplanted with the neurons may support the engraftment and differentiation of the neurons. In terms of motor function, transplantation of anterior horn cells of the spinal cord into peripheral nerves is currently superior to transplantation of differentiation-induced cells. Therefore, in this study, we attempted to transplant porcine neural cells with a view to clinical application.

Reviewer 3 Report
The chosen topic is an interesting and actual one, especially since in the case of lower motor neuron diseases there is no cure yet. The aim of this study, to investigate whether porcine fetal neural stem cells can engraft in rats peripheral nerves and regenerate functional neuromuscular junctions, was well done. In further studies it would be good that authors investigate the myelin sheath, what happens to Schwann cells during nerve regeneration, what does the myelin sheath look like during nerve regeneration. Being a peripheral nerve, I consider that is an important aspect to follow.
I would still have some remarks.
It will be necessary to present a diagram showing exactly the surgical procedure for preparing the peroneal nerve of rats for cell transplantation and how the cells were injected / where was the cell transplant site. Did the proximal and distal portions of the nerve remained ligated and only the implanted cells established the connection with the muscle? However, in figure 7 the cross section looks like a well-formed nerve, with several fascicles, with well-defined perineurium, endoneurium and myelinated axons with different sizes, even if the myelin sheath is thin in most of them.
In Figure 7 it is necessary to present a cross-section image for each group, including the control. The same for the longitudinal sections. It is not relevant to present only a cross section and a longitudinal one, from different groups. It must be clearly seen a difference between groups, on the same type of sections. It would also be good to make myelinated fiber size histograms (percentage of fibers (%) for each fiber diameter category) to assess the proportion of fibers of different sizes for each group.
Author Response
Point 1: The chosen topic is an interesting and actual one, especially since in the case of lower motor neuron diseases there is no cure yet. The aim of this study, to investigate whether porcine fetal neural stem cells can engraft in rats peripheral nerves and regenerate functional neuromuscular junctions, was well done. In further studies it would be good that authors investigate the myelin sheath, what happens to Schwann cells during nerve regeneration, what does the myelin sheath look like during nerve regeneration. Being a peripheral nerve, I consider that is an important aspect to follow.
I would still have some remarks.
It will be necessary to present a diagram showing exactly the surgical procedure for preparing the peroneal nerve of rats for cell transplantation and how the cells were injected / where was the cell transplant site. Did the proximal and distal portions of the nerve remained ligated and only the implanted cells established the connection with the muscle? However, in figure 7 the cross section looks like a well-formed nerve, with several fascicles, with well-defined perineurium, endoneurium and myelinated axons with different sizes, even if the myelin sheath is thin in most of them.
Response 1:
Thank you for your insights and suggestions to improve the manuscript. We resected the peroneal nerve a week before transplantation to prevent nerve regeneration via proximal axonal extension. The distal nerve stump and proximal nerve stump were ligated. And, proximal nerve stump was inverted proximally, and sutured into the gluteus muscle. At the time of peripheral nerve harvesting for tissue evaluation, the distal stump formed ganglia, but there was no continuity of the peripheral nerve from the proximal side. And, no regenerative axons were observed in the control group. As suggested, we have added a diagram in Figure 8 to show the surgical procedure used to prepare the peroneal nerve of rats and inject the medium containing neural cells.
Point 2:
In Figure 7 it is necessary to present a cross-section image for each group, including the control. The same for the longitudinal sections. It is not relevant to present only a cross section and a longitudinal one, from different groups. It must be clearly seen a difference between groups, on the same type of sections. It would also be good to make myelinated fiber size histograms (percentage of fibers (%) for each fiber diameter category) to assess the proportion of fibers of different sizes for each group.
Response 2:
As you pointed out, it is necessary to present a cross-section image for each group including the control. Therefore, we added cross-section images for each group in Figure 7. However, because we made all the sections into cross sections instead of longitudinal sections, we cannot present image of the longitudinal sections. We also added histograms depicting the size of the myelinated fibers in Figure 7. The distribution of myelinated axon diameter in each group did not differ among the groups.
The involvement of endogenous Schwann cells and transplanted glial cells in myelination and neuronal support is not well understood. However, our previous studies in rats suggest that both peripheral nervous system and central nervous system cells are involved in myelination of regenerating axons by electron microscopy [36]. We should clarify the process of myelination of regenerating axons in future studies.
Round 2
Reviewer 1 Report
The authors replied to my comments. I believe the manuscript is improved.